# Emerging Trends and Innovation Modes of Internet Finance—Results from Co-Word and Co-Citation Networks

**Xiaoyu Li [1], Jiahong Yuan [1], Yan Shi [2], Zilai Sun [1] and Junhu Ruan [1,\*]** 

[1] College of Economics and Management, Northwest A&F University, Yangling 712100, China; lxy2018910@nwafu.edu.cn (X.L.); yjh51251@nwafu.edu.cn (J.Y.); drszl@nwafu.edu.cn (Z.S.)

[2] Center for Liberal Arts, Tokai University, Kumamoto 862-8652, Japan; yshi@ktmail.tokai-u.jp

\* Correspondence: rjh@nwafu.edu.cn; Tel.: +86-1850-2924-249

**Abstract:** Internet finance is a financial mode combining traditional financial industry with Internet technologies, which has become a crucial part of the financial field. Due to the rapid change of information technologies and public financial needs, Internet finance has produced quite a few specific operation modes, which have interested many scholars. To better appreciate its development process and innovation modes, we used bibliometrics to analyze 2,877 articles on Internet finance in Web of Science. Through the co-word network, co-citation network and various results generated by CiteSpace, we recognized six main modes of Internet finance, that is, Internet bank, peer to peer lending (P2P lending), crowdfunding, big data finance, digital currency and fintech. Emerging research topics and the development history of each mode are also detected. We find that the mainstream modes in current research are P2P lending and crowdfunding and the research on fintech and digital currency has just begun. Through the review, we also suggest some research directions for the research direction of each mode. These results will help to deepen relevant scholars' understanding of Internet finance and provide guidance for them to choose research directions.

**Keywords:** internet finance; emerging trends; bibliometrics; Co-word analysis; Co-citation analysis

## 1. Introduction

Internet finance is a kind of financial mode that relies on Internet tools to realize financing and information exchange [1]. Traditional financial modes are mainly indirect financing represented by commercial bank and direct financing represented by investment banks. However, as a combination of Internet tools and finance, Internet finance is quite different from the traditional financial modes. On the one hand, modern information technologies, especially mobile payment, search engines and cloud computing, have a fundamental impact on human financial behavior. On the other hand, the allocation of financial resources is being changed by Internet finance [2]. Internet finance has greatly reshaped existing financial systems, such as commercial banks [3], capital market [4] and financial regulation [5]. At the same time, due to the diversification of information technologies and financial needs, Internet finance has given birth to many specific financial modes. Luo defined six Internet finance modes, namely third-party payment, peer to peer lending (P2P lending), big data finance, crowdfunding, information-based financial institution and Internet financial portal [6]. However, the inherent spirit of the Internet such as openness, equality and sharing constantly impacts the rigorous operation mode of finance. And Internet finance is constantly being combined with emerging technologies to form various new modes. Thus, it is necessary to keep a close eye on the development of the Internet finance modes, which is also important to financial supervision and macro-control [1].

Early summary research on Internet finance mainly focused on Internet banking. Hanafizadeh et al. analyzed the adoption of Internet Bank from the perspective of information technology, finance, marketing and service management [7]. After that, the summary research on Internet finance began to diversify. Martinez-Climent et al. analyzed literature related to crowdfunding [8]. Huang and Zhao studied the impact of crowdfunding on financial regulation [9]. Agarwal et al. reviewed the researches concerned with the influence of network information on financial market [10]. Cai discussed the impact of crowdfunding and blockchain on traditional financial intermediation [11]. Milian et al. examined key themes and trends in fintech [12]. It can be seen that most of the existing summary studies focused on one specific mode of Internet finance. Few papers simultaneously examine various modes of Internet finance, which is insufficient for a comprehensive understanding of the development of Internet finance.

Since research fronts in a given field refer to the body of studies that scientists actively cite in the field [13], we used CiteSpace, a citation visualization analysis software, to analyze literature of Internet finance to detect the emerging trends and innovation modes. We used the co-word network to detect various modes generated in the development process of Internet finance. Six main modes are recognized, that is, Internet bank, P2P lending, crowdfunding, big data finance, digital currency and fintech. Then, through the generated co-citation network, we analyzed the main research topics and duration of each mode and the latest research hotspots.

This paper is arranged as follows—firstly, materials and methods are introduced in Section 2, co-word analysis is carried out in Section 3 and co-citation analysis is presented in Section 4. In Section 5, we summarize our conclusions and put forward some suggestions for future research.

## 2. Materials and Methods

### 2.1. Data Source

Since emerging trends and cooperation networks in a field are observable in the dynamics of research fronts [13], we need to find appropriate literature on Internet finance. Web of Science (WoS) Core Collection includes the world's leading scholarly journals and proceedings in the sciences and social sciences and navigates the full citation network. Therefore, we searched the key words about Internet finance from the core collection database of WoS to obtain article records. One issue worth noting here is the accuracy and breadth of our search for literature. The more literature we get, the more likely we are to add irrelevant literature. But when we carefully screen, the amount of literature we obtain may not be enough to cover all the fields of Internet finance. According to Chen, the CiteSpace software developer, unwanted documents can be removed by the software itself [14], so we sacrifice accuracy for breadth.

The data we used are the comprehensive WoS search results, which is shown in Table 1. First, we need to confirm that topics about Internet finance are included in the data. And we get results #1, #2 and #3 by setting keywords such as Internet bank, Internet banking and Internet finance and combine them to result #4. After that, we set the time from 2008 to 2018 and get the result #5. Because we mainly need literature on the development modes of Internet finance, we need to eliminate the literature related to pure technologies. Therefore, we get result #6 by setting subject categories as economics, business, business finance and other related disciplines. Then, we set document type as article to get result #7 and finally get 2877 results for analysis.

### 2.2. Citespace

Bibliometrics was proposed by American bibliographer Alan Pritchard in 1969, whose main content includes co-citation analysis, co-author analysis, co-word analysis and so on. Co-citation analysis was initially proposed by Small in 1973 [15] and has been recognized as one effective tool to detect research fronts, intellectual bases and development trends in the scientific literature [14]. Co-word analysis was first proposed by Serge Bauin in the 1980s, which was a way to confirm the

relationship between topics in the subject area represented by the text by analyzing the co-occurring forms such as keywords in the same text [16]. Therefore, this article uses these two types as analysis tools. Many softwares can be used for bibliometric analysis, among which the CiteSpace software developed by Professor Chen is scientifically effective, easy to use [13]. Co-citation analysis theory and co-word analysis theory are the foundation of it, so we use CiteSpace to detect the main modes and development of Internet finance.

**Table 1.** The search results of Internet finance in Web of Science (WoS).

| Number | Number of Records | Search Settings |
| :---: | :---: | :---: |
| #7 | 2877 | #6 Refined by: DOCUMENT TYPES: (Article) |
| #6 | 2990 | #5 Refined by: WoS CATEGORIES: (Management OR Business OR Economics OR Business Finance OR Communication) |
| #5 | 8910 | #4 Refined by: PUBLICATION YEARS (2008-2018) |
| #4 | 10743 | #3 OR #2 OR #l |
| #3 | 3615 | TS = (Internet Bank OR Online Bank OR Electronic Bank OR E-Bank OR Internet-Based Bank) |
| #2 | 3615 | TS = (Internet Banking OR Online Banking OR Electronic Banking OR E-Banking OR Internet-Based Banking) |
| #1 | 7649 | TS = (Internet Financ* OR Online Financ* OR Electronic Finance* OR E-Financ* OR Internet-Based Financ*) |

In order to make our results easy to understand, we briefly introduce the generation principle and some basic concepts of CiteSpace. (1) Co-word analysis. Each article will cite some keywords, which indicates that these keywords have a certain relationship [17]. According to this principle, CiteSpace generates networks of keywords to show these relationships. (2) Co-citation analysis. If two works are cited in the same paper, it shows that they may have a certain correlation [14]. Furthermore, the more the two works are cited at the same time, the stronger the correlation between the two references will be. CiteSpace generates a co-citation network according to the above principle. (3) Cluster labelling algorithms. Cluster labelling algorithms refer to how to label clusters in the co-citation network. The log-likelihood ratio (LLR) algorithm is recognized to effectively recognize labels with better representativeness [14]. Thus, we use the LLR algorithm in the work. (4) Indicators and reports. CiteSpace can generate various reports by indicators. Betweenness centrality proposed by Freeman, is used as the measure of centrality of one document in citation networks. Studies have shown that nodes with high betweenness centrality values tend to identify boundary spanning potentials that may lead to transformative discoveries [18]. Citation burstness is a computational technique that has been used to identify abrupt changes of events and other types of information [19]. We mainly use Explosively Cited Papers Report based on betweenness centrality and citation burstness which provides explosively cited papers sorted by year to find the latest hotspots [14]. (5) Related Literature. Citespace can provide relevant literature for each node of the generated network, which can be used for further analysis.

## 3. Co-Word Analysis

### 3.1. Main Modes of Internet Finance

We set the date from 2008 to 2018 and CiteSpace will generate a co-word network of keywords which is shown in Figure 1. The more times one keyword appears, the larger the font size in the graph and the biggest label can be considered as the main and macro research objects [14]. As can be seen from Figure 1, the main keywords detected are Internet bank, fintech, crowdfunding, digital currency and P2P lending. They must appear in the literature a lot and basically reveal the research directions of Internet finance which can be regarded as the main modes of Internet finance. As for other labels, their meanings are too macro to be interpreted in the work.

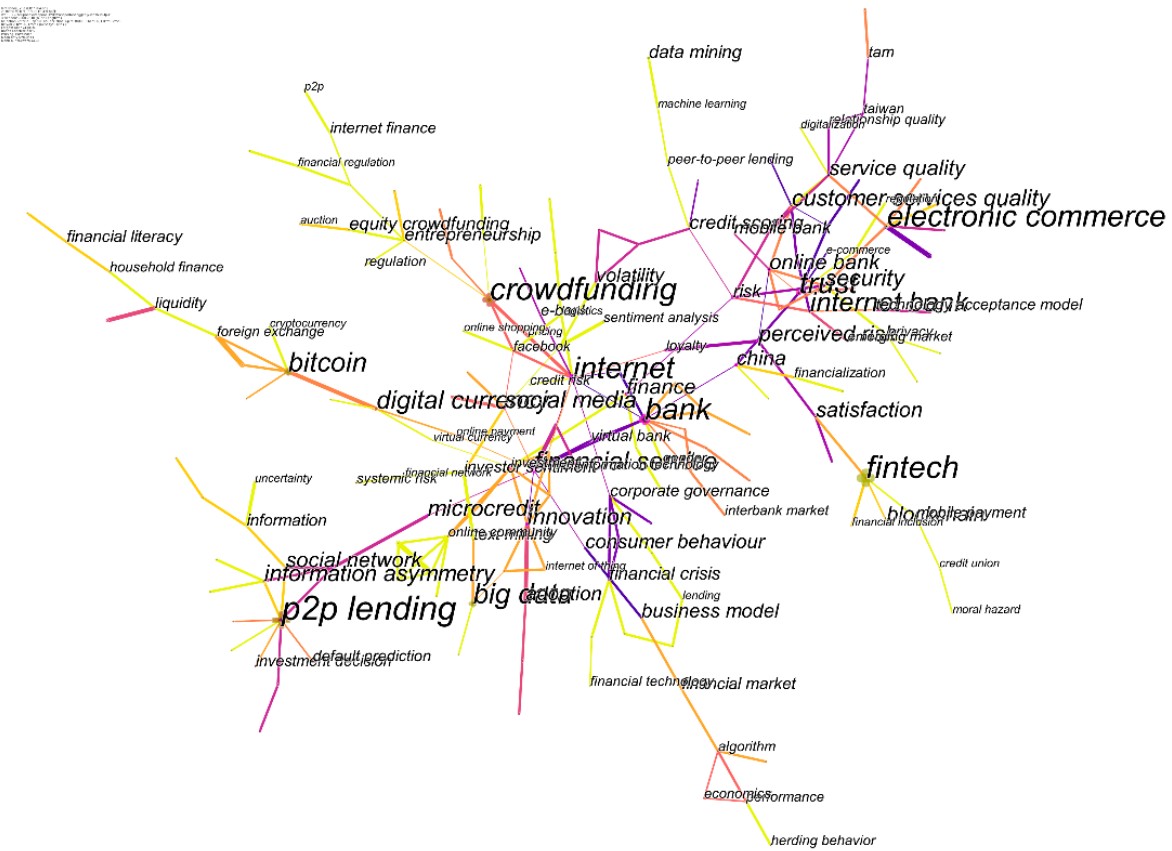

**Figure 1.** Co-word network of keywords.

The color of the line indicates the time the relevant study occurred. Blue and purple mean the study came earlier, while yellow and orange mean the opposite. As for the green and the red lines, they indicate that the study appeared in the middle period. Table 2 lists the number and color of lines connected to each mode. According to the color of their lines, both e-commerce and Internet banking have at least one blue or purple line, indicating that these modes appear early and there are also many lines of other colors connecting with them, suggesting related research is constantly expanding. For P2P lending and crowdfunding, they do not have blue or purple lines, indicating that their research is relatively new. As for fintech and digital currency, their lines are mostly yellow and orange, which means their research just starts. Besides, keywords like Internet banking, P2P lending and crowdfunding have many branches, which means they have more research topics compared with fintech and digital currency.

**Table 2.** Number and color of each pattern directly connected to them.

| Keywords | Number of Lines | | |
|---|---|---|---|
| | Blue and Purple | Red and Green | Yellow and Orange |
| Internet bank | 2 | 2 | 0 |
| E-commerce[1] | 2 | 2 | 1 |
| Crowdfunding | 0 | 3 | 2 |
| P2P lending | 0 | 4 | 4 |
| Digital currency | 0 | 0 | 4 |
| Fintech | 0 | 0 | 4 |

[1] E-commerce is closely related to Internet finance but it is not a mode of Internet finance.

### 3.2. Further Analysis of the Research Perspective

We can further analyze the main node and its secondary nodes of each mode. We only perform analysis on crowdfunding due to the complexity of the generated network, while the analysis of other modes is performed in co-citation analysis. We use a network of nodes and sub-nodes connected to crowdfunding to analyze the keyword network in detail to understand related main topics, which is shown in Figure 2. We can see the adjacent keywords connected with each keyword from the graph. Keywords linked to crowdfunding are Entrepreneurship, Screening, Mechanism design, Pricing, Facebook and Internet. Since Internet is too broad to analyze, we ignore it. Table 3 lists the detected important literature, number of citations and main content for each keyword. We can see that crowdfunding includes two types of crowdfunding—equity crowdfunding and reward-based crowdfunding. There are many perspectives to study these two types of crowdfunding, including social networking, government regulation, investor, corporate and so on. However, keywords cannot represent the whole research content of each mode and we will analyze the development and topics of Internet finance modes through co-citation network.

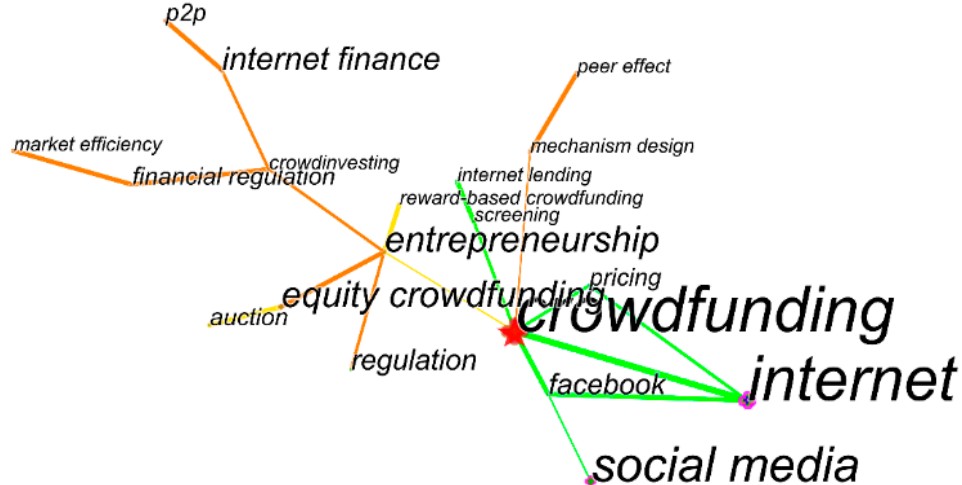

**Figure 2.** Keywords co-word network of crowdfunding.

**Table 3.** The main information of nodes in network of crowdfunding.

| Keywords | Content | Reference | Citations[1] |
|---|---|---|---|
| Facebook | The influence of social network on crowdfunding | [20] | 48 |
| Pricing | Pricing of crowdfunding advertisement | [21] | 20 |
| Auction | The role of auction in equity-based crowdfunding | [21] | 8 |
| Regulation | Regulation of cross-border crowdfunding | [22] | 6 |
| Equity crowdfunding | The impact of market regulation and agency risk | [23] | 4 |
| Entrepreneurship | The influence of moral ethics on crowdfunding | [24] | 4 |
| Mechanism design | Key point of designing crowdfunding mechanism | [25] | 1 |
| Reward-based Crowdfunding | Factors influencing the success of crowdfunding | [26] | 1 |
| Financial regulation | Difference between crowdfunding and illegal fundraising | [27] | 1 |

[1] represents the cited times in the 2877 documents we selected, not the real cited times.

## 4. Co-Citation Analysis

### 4.1. Main Topics of Each Mode

Using CiteSpace, we detected 10 clusters from the 2877 articles and their 11003 references, as shown in Figure 3 (cluster *#9* is not visible due to the display threshold setting of cluster size). CiteSpace automatically generates tag names from keywords, abstracts and titles of documents according to LLR algorithm, which is mentioned in the second part of this paper. It is worth noting that the name of the cluster comes from the title, summary and keywords and does not necessarily represent the actual research content of the cluster. Table 4 lists the main works of each cluster, sorted by relevance and color, with clusters unrelated to Internet finance placed at the end. We will use the main literature of each cluster to explore the main topics of each mode.

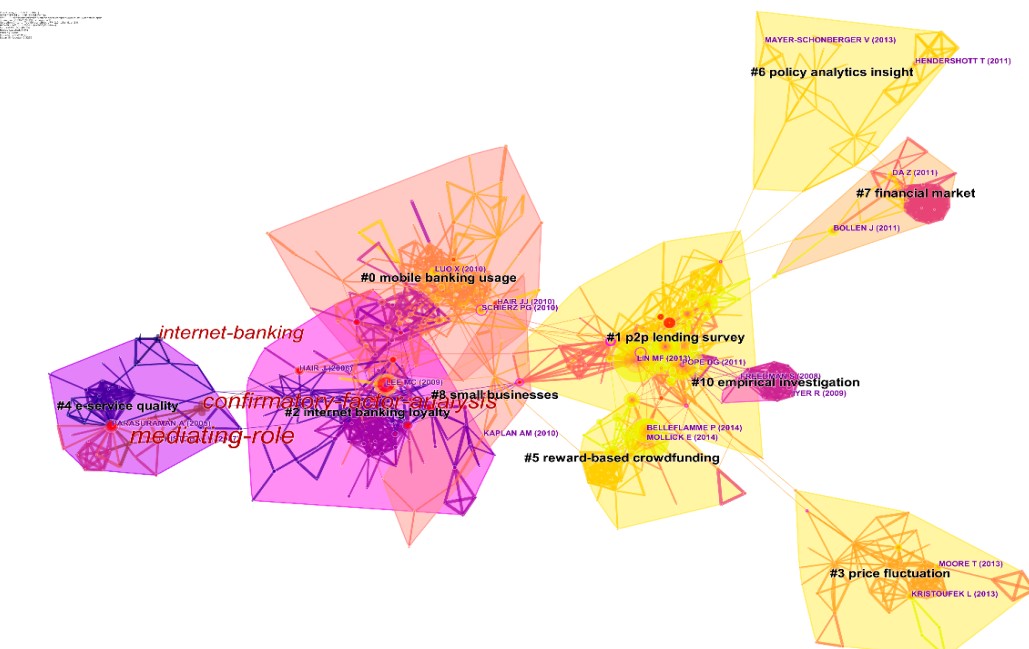

**Figure 3.** Co-citation networks.

#### 4.1.1. Internet Bank

In cluster *#2 Internet banking loyalty* and cluster *#0 Mobile banking usage*, the first two cluster list in Table 4, the research topics mainly focused on Internet bank or mobile bank. And the research content is basically similar, mainly in terms of user adoption and influencing factors of user loyalty. In terms of user adoption, both traditional and specific factors are involved. Conventional factors include perceived ease of use, perceived usefulness, trust, cognition and perceived risk [18,28–30]. Specific factors include income [18,29], personal innovation [31], consumer behavior mode [18] and system compatibility [32]. The general factors that influence user loyalty are basically same as those affecting user adoptions and factors such as satisfaction, subjective norms and attitudes are added [33,34]. Besides, A new research direction is how to make financial platforms such as platform of pension products more transparent and efficient [35]. There is no unified conclusion about the importance of various factors but Sripalawat et al. found that positive factors had more impact on the willingness to use mobile banking than negative factors [36]. At the same time, with the development of research, the considered factors expand from conventional factors to unconventional factors such as national culture [37] and system compatibility [32] and the research countries also transfer from developed countries to developing countries like Vietnam [38], Iran [32] and Thailand [36].

### 4.1.2. P2P Lending

In cluster *#1 P2P lending survey*, the main research content is P2P lending. We have analyzed the main works on it, which list in Table 4. The literature in the cluster mainly studies the influencing factors of lending behavior and highlights the limitations caused by information asymmetry. The factors that affect the success and interest rate of P2P lending mainly include trust, online friendship and social differences [39–42]. Since most investors are not professional investors, they do not have enough ability to evaluate lenders from limited information, thus showing two problems—herd effect and information asymmetry. Zhang and Liu found evidence of rational herding effect among lenders, which leads to the weakening effect of bidding mechanism [41]. At the same time, due to information asymmetry, borrowers are with an information disadvantage in terms of loan evaluation, which limits the development of P2P reception [43]. Lin et al. studied the information asymmetry in P2P lending [40]. And Zhao et al. firstly provided a systematic classification of P2P lending and made a detailed comparison of their working mechanism, which is very helpful for understanding P2P lending [44].

### 4.1.3. Crowdfunding

Cluster *#5 Reward-based Crowdfunding* is the research about crowdfunding. Crowdfunding includes product crowdfunding and equity crowdfunding and Belleflamme et al. compared the differences between the two [45]. The research on crowdfunding can be divided into two phases—crowdfunding phase and after crowdfunding phase. The first stage mainly studies the factors that affect the success of crowdfunding, including the information of the sponsor, the quality of the project, the type of the project [46], the capital of the company and detailed risk information [47]. Burtch et al. studied the herd effect in crowdfunding and found that the behavior of investors has mutual influences, which affects the results of crowdfunding [48]. The second stage mainly studies the influencing factors of satisfaction of crowdfunding results, including the timeliness of payment, compliance standards, the utilitarian value and hedonic value of the project, the timeliness of delivery, product quality and project novelty [49–51]. Cholakova and Clarysse argued that profit was the main motivation for investors to participate in crowdfunding. In addition, many scholars interpret crowdfunding from the regulatory perspective [52,53]. Liu et al. focused on the differences between crowdfunding and illegal fundraising and the supervision of illegal fundraising [27]. Zetzsche and Preiner detailed how European regulators promote a single European crowdfunding market [22].

### 4.1.4. Digital Currency

The main research object of cluster *#3 Price fluctuation* is digital currency. We have listed 6 important documents in Table 4. Boehme et al. introduced the design principles and features of the platform of digital currency to non-technical personnel [54]. Existing works mainly studied the factors in bitcoin price fluctuation and the prospect of bitcoin becoming a payment currency. One of the main research directions is the impact of investor's attention on the price fluctuation of Bitcoin and Barber has proved that individual investors are easily affected by news and stocks with abnormal trading volume [55]. Kristoufek studied the relationship between Bitcoin price volatility and Google search volume [56]. Polasik et al. found that the popularity of bitcoin, the emotions expressed in the newspaper reports on cryptocurrencies and the total transaction amount were the main factors driving the return rate of bitcoin [57]. Ciaian et al. argued that the main driver of bitcoin's drastic price fluctuations was its attractiveness index, followed by market forces, which would also hinder bitcoin from becoming a payment currency [58]. Another research direction is the formation mechanism of bitcoin's price bubble to explain the connotation of digital currency. Fry and Cheah established a set of financial bubble and collapse models on bitcoin and ripple and illustrated the essence of digital currency with them [59].

### 4.1.5. Big Data Finance

The main research object in cluster *#7 financial market* is big data finance. Its main research topic is obtaining information from social networks and so forth, to assist in speculating the trend of the asset market. Bollen et al. found that specific dimensions of public emotion extracted from social networks can significantly improve the accuracy of Dow Jones Industrial Average prediction [60]. Da et al. proposed a new method to directly measure investor attention by using Google search frequency [61]. Preis et al. analyzed the changes of financial related search terms in the Google query volume and found that the pattern of "early warning signal" that can be interpreted as stock market volatility [62]. Kyriakou et al. studied the application of machine learning in stock return research [63]. Moat et al. found that the data on the change of page views of Wikipedia's financial pages contained the early signs of the stock market trend, so the online data were useful in the information collection stage before the decision [64].

**Table 4.** Main works of each cluster.

| Cluster ID | Cited Reference | | | Citing Reference | | |
|---|---|---|---|---|---|---|
| | Cites[1] | Reference | Year | Coverage[2] | Reference | Year |
| *#0 Mobile bank usage* | 13 | [28] | 2009 | 17 | [32] | 2015 |
| | 13 | [65] | 2010 | 12 | [36] | 2011 |
| | 12 | [18] | 2009 | 11 | [37] | 2015 |
| *#2 Internet banking loyalty* | 31 | [29] | 2009 | 20 | [33] | 2011 |
| | 14 | [30] | 2003 | 19 | [34] | 2011 |
| | 14 | [31] | 2007 | 14 | [66] | 2011 |
| *#1 P2P lending survey* | 56 | [39] | 2013 | 18 | [44] | 2017 |
| | 41 | [40] | 2011 | 12 | [67] | 2015 |
| | 36 | [41] | 2012 | 9 | [42] | 2014 |
| *#5 Reward-based Crowdfunding* | 53 | [46] | 2014 | 30 | [25] | 2017 |
| | 34 | [45] | 2014 | 18 | [50] | 2017 |
| | 20 | [48] | 2013 | 6 | [52] | 2015 |
| *#3 Price fluctuation* | 11 | [56] | 2013 | 30 | [57] | 2016 |
| | 10 | [54] | 2013 | 28 | [58] | 2016 |
| | 9 | [55] | 2015 | 4 | [59] | 2106 |
| *#7 Financial market.* | 16 | [60] | 2011 | 30 | [62] | 2013 |
| | 13 | [61] | 2011 | 22 | [64] | 2013 |
| *#4 E-service quality* | 17 | [68] | 2005 | 30 | [69] | 2009 |
| *#6 Policy analytics insight* | 8 | [70] | 2011 | 26 | [71] | 2017 |
| *#8 Small business* | 13 | [72] | 2010 | 26 | [73] | 2014 |
| *#10 Empirical investigation* | 33 | [74] | 2012 | 6 | [75] | 2008 |

[1] represents the cited times in the its cluster we selected, not the real cited times. [2] is the ratio of literature cited to all literature in its cluster.

### 4.1.6. Others

*#4 E-service quality, #6 Policy analytics insight, #8 small business* and *#10 empirical investigation* are irrelevant to Internet finance. So, we only list one most cited work and one most 'Citing' work in Table 4. We can see that big data finance does not appear in the co-word network of keywords explicitly but it appears in the co-citation network as a cluster, indicating that it is an important mode of Internet finance. Meanwhile, fintech does not appear in the co-citation network, which may indicate that it appears late and has not yet formed a cluster. This result indicates that simultaneous analysis of co-word and co-citation can avoid missing some important result.

### 4.2. Duration of Each Mode

The time-line form of the co-cited network is shown in Figure 4, which is just another manifestation of co-citation network and is more helpful for us to judge research hotspots and duration of each mode.

The dot on the horizontal line indicates a document in the cluster in which it belongs and the connection between the documents indicates their co-citation relationship. On the right are the clusters which the reference belongs to and the corresponding timeline coordinates represent the year of publication. Because we set that literature can only appear in the network if it is cited more than a certain number of times, some clusters which have not lasted until now, do not mean that there is no latest literature but their influence is less. Therefore, when the clusters have not lasted until now, there are two possibilities here. One possibility is that relevant research is no longer a research hotspot and the latest research is ignored, which is shown in Figure 4 as clusters #0, #2 and #4. Another possibility is that the relevant researches have just started and there are fewer networks formed, which is shown in Figure 4 as clusters #3, #7 and #10. The clusters that last until now are #1 and #4.

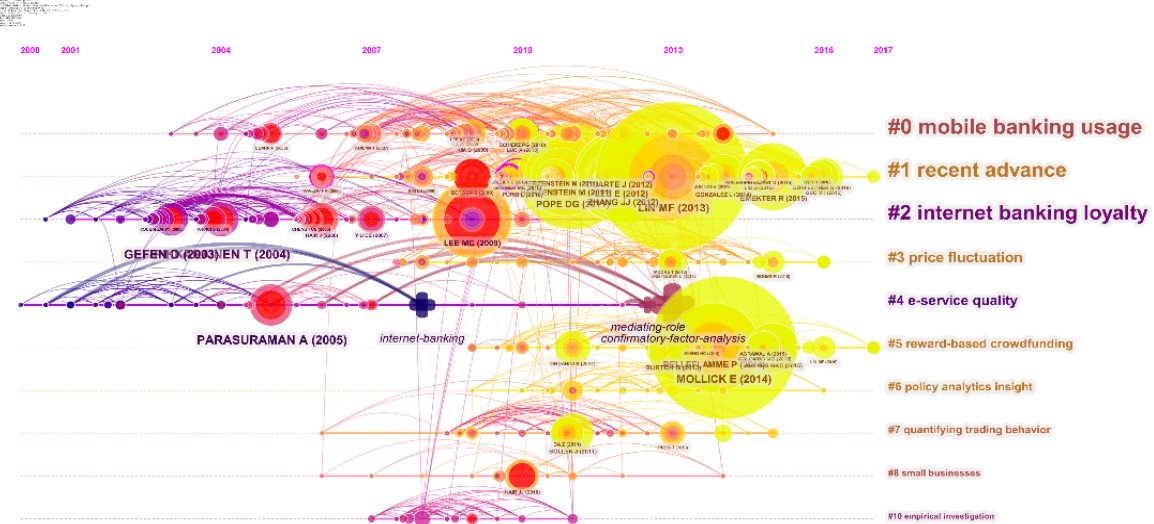

**Figure 4.** Time-line form of the co-cited network.

We have discussed the research directions that each cluster represents in Section 4.1. Therefore, we can divide the modes of Internet finance into three categories. The first category consists of Internet banking and mobile bank and their related researches are no longer the current hot spot. The second category consists of crowdfunding and P2P lending and their works are the latest mainstream. The third category includes modes like big data finance and digital currency, which are just started. There may be other patterns but fail to show up because they are not related enough with other studies.

### 4.3. The Latest Research Hotspot

Reports generated by CiteSpace provide explosively cited papers sorted by year of termination, which is shown in Table 5. The value of strength calculated by CiteSpace refers to the strength of citation bursts and red lines indicate the duration of the outbreak in the last column [13,14]. If the number of citations of a reference continues to surge until now, it may indicate that its main research topics have been repeatedly studied and its topics are recent hotspots [18]. We can judge recent research hotspots through the interpretation of these papers. Berger and Gleisner found that financial intermediaries in the P2P lending market could significantly reduce the problem of information asymmetry [76]. Liu et al. analyzed that in P2P lending, friend relationship could not only help the borrower to raise funds but also produce herding effect [77]. As for the rest of the literature, their citation explosion ended, indicating that the relevant hot spot has ended. We find an ongoing research hotspot, that is, P2P lending. Information asymmetry and herd behavior in P2P lending have appeared many a time in related papers, which is a very important research hotspot in the field of Internet finance.

**Table 5.** Main information of explosively cited papers sorted by year of termination.

| Reference | Year | Strength[1] | Begin | End | 2008–2018[2] |
|:---:|:---:|:---:|:---:|:---:|:---:|
| [76] | 2009 | 3.4235 | 2015 | 2018 | |
| [78] | 2015 | 2.7834 | 2016 | 2018 | |
| [79,80] | 2010 | 4.3255 | 2014 | 2016 | |
| [29] | 2009 | 3.5534 | 2015 | 2016 | |
| [77] | 2014 | 3.2422 | 2015 | 2016 | |

[1] Indicates the intensity of the citations cited in the literature [14]. [2] The cyan line indicates the continuous year after the publication of the document and the red line indicates the continuous year of the document's explosive citation.

## 5. Major Findings and Outlook

### 5.1. Major Findings

From our findings above, we can see that Internet finance modes include Internet banking, crowdfunding, P2P lending, big data finance, digital currency and fintech. And we divide them into three stages according to their development level. The first stage is Internet banking. The second stage includes modes like P2P lending and crowdfunding and the third stage includes modes like fintech, big data finance and digital currency. In addition, we found that during literature analysis, simultaneous co-word analysis and co-citation analysis can avoid missing important results to a certain extent.

- Internet banking. Research related to Internet banking is the earliest. In the early days, it mainly studied the influence of various factors such as risk perception, perceived benefits, attitudes and perceived usefulness on user adoption of Internet bank. Later studies focused on the impact of factors such as trust, security, authentication and customer service quality on user loyalty and user usage depth of Internet bank and the research goal turned to the use of mobile banking. The research angle expands from the perspective of the earliest interests and trust to national culture, from the perspective of developed countries to developing countries.

- P2P lending and crowdfunding. The researches of P2P lending and crowdfunding are still developing, which still have potential to explore. And research hotspot focused on these modes. In terms of P2P lending, the content of its research is related to P2P lending success factors like ethnic differences, information asymmetry, herd effect and default prediction. And the main research content is information asymmetry, which is also a research hotspot. As for crowdfunding, one main research content refers to determinants of project crowdfunding success such as information on crowdfunding project sponsors, the quality of crowdfunding projects, the types of crowdfunding projects and investor satisfaction and donors' behavior. Other research contents include the design of crowdfunding mechanism, the influence of social network on crowdfunding and the use of equity crowdfunding or product crowdfunding. We can study crowdfunding from the perspective of investors, regulators and sponsors.

- Big data finance, fintech and digital currency. The third stage involves fintech, big data finance and digital currency. Their research content is small but constantly enriched and developed. The rapid development of bitcoin has brought the research upsurge of digital currency. Its essence is the main research direction of price decision-making mechanism. The current research on fintech mainly is using big data to predict changes in asset market.

### 5.2. Outlook

Based on the results of our analysis, we propose some views on future Internet finance research.

- Information asymmetry in P2P lending is the latest research hotspot and it will continue to develop. The emergence of big data related technologies may solve this problem.

- There will be new research hotspots in the field of crowdfunding. For example, crowdfunding as a financing method is naturally suitable for microfinance and therefore has great help for China's

rural poverty alleviation. But because of the risk of crowdfunding, if there is corresponding crowdfunding insurance, then crowdfunding may enter a stage of rapid development. Of course, the pricing of crowdfunding insurance is also very difficult and requires further study.

- Fintech is still in the first stage and will also develop its own research content, such as artificial intelligence assisted analysis and forecasting market in financial technology, inclusive finance and supervision of emerging content. Artificial intelligence analysis of financial market trends has broad prospects. It can use tens of thousands of indicators to predict capital markets such as stock markets by training automatic screening indicators. And it can be combined with big data to obtain indicators that are difficult to quantify, such as investor sentiment and consumer sentiment, to make predictions more accurate.

- Big data finance can help analyze the credit status of individuals. The combination of big data and artificial intelligence can accurately judge the credit status of individuals and the judgment of each individual will form the credit status of groups, enterprises, regions and countries. And the impact of macroeconomic policies on individuals can be visualized, helping us to make better investment decisions and government policies.

- Digital currency can reduce transaction costs but needs to avoid risks. Digital currency may break a country's limits in the way it consumes a lot of power before it understands it needs to change. In addition, due to the borderless nature of digital currency, we also need to constantly update the regulatory approach.

## 6. Conclusions

Internet finance has increasingly penetrated into our lives, bringing changes in the allocation of financial resources and human financial behavior. With the development of Internet technology and the diversity of financial needs, various new modes of Internet finance appear. Hence, it is important to understand the various modes generated in the development process of Internet finance and main research topics and duration of each mode of them. Different from the existing summary studies, which focused on one specific mode of Internet finance [7–12], we used bibliometrics to simultaneously examine various modes of Internet finance. First, we carried out a co-word analysis of the literature, which showed the main modes of Internet finance. Six main modes were recognized, that was, Internet bank, P2P lending, crowdfunding, big data finance, digital currency and fintech. And Emerging research topics and the development history of each mode were also detected through the co-citation network. Additionally, the mainstream modes in current research were also identified.

**Author Contributions:** Conceptualization, X.L. and J.R.; methodology, X.L. and J.R.; software, X.L.; formal analysis, X.L. and J.Y.; resources, X.L. and J.Y.; data curation, X.L.; writing—original draft preparation, X.L. and J.Y.; writing—J.R. and Y.S.; visualization, X.L. and J.Y.; supervision, Z.S., J.R. and Y.S.; funding acquisition, J.R.. All authors have read and agreed to the published version of the manuscript.

**Funding:** This research is supported by the National Natural Science Foundation of China (71703122, 71973106) and the National Key Research and Development Program of China (2019YFD1101103).

**Conflicts of Interest:** The authors declare no conflict of interest.

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
