# Peer review of "Emerging Trends and Innovation Modes of Internet Finance—Results from Co-Word and Co-Citation Networks"

_futureinternet, doi:10.3390/fi12030052_

Round 1

Reviewer 1 Report

The topic of the manuscript is relevant to the scope of Future internet.

The paper is well structured and written. The title of the article is clear and adequate. The abstract is clear, it presents the object of research the content and the results.  The introduction states the objectives of the paper.  The methodology seems sound.

The results and interpretations are correct. 

The authors highlighted the policy implications of their findings.

Author Response

Response to Reviewer 1

Manuscript ID: futureinternet-746692

Title: Emerging trends and innovation modes of Internet finance: Results from co-word and co-citation networks

12 March 2020

Dear reviewer,

We would like to thank you for the helpful comments on our paper. Here are our responses to the comments. The comments are numbered and in italics.

Thanks for your attention and my best regards.

Xiaoyu Li, Jiahong Yuan, Yan Shi, Zilai Sun and Junhu Ruan.

Point 1: The topic of the manuscript is relevant to the scope of Future internet.

The paper is well structured and written. The title of the article is clear and adequate. The abstract is clear, it presents the object of research the content and the results. The introduction states the objectives of the paper. The methodology seems sound.

The results and interpretations are correct.

The authors highlighted the policy implications of their findings.

Response 1: Thanks for your confirmation on our contribution and constructive comments. We have revised our work strictly based on your suggestions.

Point 2: English language and style are fine/minor spell check required。

Response 2: Yes. You are right. We have made the following adjustments:

  1. In section 2.2, the previous statement “Log-likelihood ratio (LLR) algorithm is recognized to effectively recognize labels with better representativeness [14].” has been changed into “The log-likelihood ratio (LLR) algorithm is recognized to effectively recognize labels with better representativeness [14].”
  2. In section 3.2, the previous statement “While Internet is too board to analyze, we ignore it.” has been changed into “Since Internet is too broad to analyze, we ignore it.”
  3. In section 3.2, the previous statement “There may be other patterns, but fail to show up because of not with enough related studies.” has changed into “There may be other patterns, but fail to show up because they are not related enough with other studies.”
  4. In introduction, the previous statements “Hanafizadeh”, “Martinez-Climent”, “Huang”, “Agarwal” and “Milian” have been changed into “Hanafizadeh et al”, “Martinez-Climent et al.”, “Huang and Zhao”, “Agarwal et al.” and “Milian et al.”.
  5. In section 4.1.1, the previous statement “Sripalawat” has been changed into “Sripalawat et al”.
  6. In section 4.1.2, the previous statements “Zhang”, “Lin” and “Zhao” have been changed into “Zhang and Liu”, “Lin et al.” and “Zhao et al.”.
  7. In section 4.1.3, the previous statements “Belleflamme”, “Burtch”, “Cholakova”, “Liu” and “Zetzsche” have been changed into “Belleflamme et al.”, “Burtch et al.”, “Cholakova and Clarysse”, “Liu et al.” and “Zetzsche et al.”.
  8. In section 4.1.4, the previous statements “Boehme”, “Polasik”, “Ciaian” and ” Fry” have been changed into “Boehme et al.”, “Polasik et al.”, “Ciaian et al.” and ” Fry and Cheah”.
  9. In section 4.1.5, the previous statements “Bollen”, “Preis”, “Da” and “Moat” have been changed into “Bollen et al.”, “Preis et al.”, “Da et al.” and “Moat et al.”.
  10. In section 4.3, the previous statements “Berger” and “Liu” have been changed into “Berger and Gleisner” and “liu et al.”

Reviewer 2 Report

The article offers contributions to the understanding of current research on internet finance, using networks of co-word and co-citation.

Although it does not provide sizeable innovation in terms of new theories or techniques, it does offer a panorama of the current state of internet finance that will be useful to researchers, so I think it does fall within the scope of Future Internet.

I have just two minor issues that I think should be addressed by the authors.

1) Articles with more than two authors should be cited as "Author 1 et al", as in "Preis et al". Articles with two authors should be cited with both surnames.

2) In Section 5, Conclusions and outlook, section 5.1. Conclusion is not truly a conclusion, but a summary of the article. Maybe the authors could consider changing the name of the section (this is a suggestion, not mandatory).

Some minor English mistakes. Please change, according to the authors' discretion:

  • “log-likelihood ratio” to “The log-likelihood ratio”;
  • “While Internet is too board” to “Since Internet is too broad”;
  • “because of not with enough related studies” to “they are not related enough with other studies”.

Author Response

Response to Reviewer 2

Manuscript ID: futureinternet-746692

Title: Emerging trends and innovation modes of Internet finance: Results from co-word and co-citation networks

12 March 2020

Dear reviewer,

We would like to thank you for the helpful comments on our paper. Here are our responses to the comments. The comments are numbered and in italics.

Thanks for your attention and my best regards

Xiaoyu Li, Jiahong Yuan, Yan Shi, Zilai Sun and Junhu Ruan.

Point 1: The article offers contributions to the understanding of current research on internet finance, using networks of co-word and co-citation. Although it does not provide sizeable innovation in terms of new theories or techniques, it does offer a panorama of the current state of internet finance that will be useful to researchers, so I think it does fall within the scope of Future Internet.

Response 1: Thanks for your confirmation on our contribution and constructive comments. We have revised our work strictly based on your suggestions.

Point 2: I have just two minor issues that I think should be addressed by the authors.

1) Articles with more than two authors should be cited as "Author 1 et al", as in "Preis et al". Articles with two authors should be cited with both surnames.

Response 2: Yes. You are right. We have made the following adjustments:

  1. In introduction, the previous statements “Hanafizadeh”, “Martinez-Climent”, “Huang”, “Agarwal” and “Milian” have been changed into “Hanafizadeh et al”, “Martinez-Climent et al.”, “Huang and Zhao”, “Agarwal et al.” and “Milian et al.”.
  2. In section 4.1.1, the previous statement “Sripalawat” has been changed into “Sripalawat et al”.
  3. In section 4.1.2, the previous statements “Zhang”, “Lin” and “Zhao” have been changed into “Zhang and Liu”, “Lin et al.” and “Zhao et al.”.
  4. In section 4.1.3, the previous statements “Belleflamme”, “Burtch”, “Cholakova”, “Liu” and “Zetzsche” have been changed into “Belleflamme et al.”, “Burtch et al.”, “Cholakova and Clarysse”, “Liu et al.” and “Zetzsche et al.”.
  5. In section 4.1.4, the previous statements “Boehme”, “Polasik”, “Ciaian” and ” Fry” have been changed into “Boehme et al.”, “Polasik et al.”, “Ciaian et al.” and ” Fry and Cheah”.
  6. In section 4.1.5, the previous statements “Bollen”, “Preis”, “Da” and “Moat” have been changed into “Bollen et al.”, “Preis et al.”, “Da et al.” and “Moat et al.”.
  7. In section 4.3, the previous statements “Berger” and “Liu” have been changed into “Berger and Gleisner” and “liu et al.”

Point 3:2) In Section 5, Conclusions and outlook, section 5.1. Conclusion is not truly a conclusion, but a summary of the article. Maybe the authors could consider changing the name of the section (this is a suggestion, not mandatory).

Response 3: Yes. You are right. We have made the following adjustments:

  1. In section 5, the previous title of section 5 “Conclusion and outlook” has been changed into “Major Findings and Outlook”.
  2. We added section 6, “Conclusion”, and in this section, we added a new conclusion: “Internet finance has increasingly penetrated into our lives, bringing changes in the allocation of financial resources and human financial behavior. With the development of Internet technology and the diversity of financial needs, various new modes of Internet finance appear. Hence, it is important to understand the various modes generated in the development process of Internet finance and main research topics and duration of each mode of them. Different from the existing summary studies, which focused on one specific mode of Internet finance [7-12], we used bibliometrics to simultaneously examine various modes of Internet finance. First, we carried out a co-word analysis of the literature, which showed the main modes of Internet finance. Six main modes were recognized, that was, Internet bank, P2P lending, crowdfunding, big data finance, digital currency and fintech. And Emerging research topics and the development history of each mode were also detected through the co-citation network. Additionally, the mainstream modes in current research were also identified.”

Point 4: Some minor English mistakes. Please change, according to the authors' discretion:

“log-likelihood ratio” to “The log-likelihood ratio”;

“While Internet is too board” to “Since Internet is too broad”;

“because of not with enough related studies” to “they are not related enough with other studies”.

Response 4: Yes. You are right. We have made the following adjustments:

  1. In section 2.2, the previous statement “Log-likelihood ratio (LLR) algorithm is recognized to effectively recognize labels with better representativeness [14].” has been changed into “The log-likelihood ratio (LLR) algorithm is recognized to effectively recognize labels with better representativeness [14].”
  2. In section 3.2, the previous statement “While Internet is too board to analyze, we ignore it.” has been changed into “Since Internet is too broad to analyze, we ignore it.”
  3. In section 3.2, the previous statement “There may be other patterns, but fail to show up because of not with enough related studies.” has been changed into “There may be other patterns, but fail to show up because they are not related enough with other studies.”

Reviewer 3 Report

The authors have written an interesting but very high-level paper on the exploding amount of research on new digital stuff in finance. I like the paper and think it should be published. I do not have much to add, except being a little more concrete a couple of places might be worth it. For example around section 4.1.1 on internet banking , one could cite recent advances on understand risk fast in a fintech environment via the new fintech methodology pointed out in 

Communication and personal selection of pension saver's financial risk  Gerrard, R.Hiabu, M.Kyriakou, I.Nielsen, J.P. 2019 European Journal of Operational Research

and in your section 4.1.5  on stock prediction, while the cited paper might be fun, then a more serious paper predicting stocks for the long-run  - that are more important for the man-on-the-street looking for long-term pension investment - and also using machine learning type of methods can be found in 

Forecasting benchmarks of long-term stock returns via machine learning 
Open Access
Kyriakou, I.Mousavi, P.Nielsen, J.P.Scholz, M. 2019 Annals of Operations Research

There are of course many more studies relevant for the current research. However, I also understand that there will have to be some limit to many specific cases that can be treated.

Author Response

Response to Reviewer 3

Manuscript ID: futureinternet-746692

Title: Emerging trends and innovation modes of Internet finance: Results from co-word and co-citation networks

12 March 2020

Dear reviewer,

We would like to thank you for the helpful comments on our paper. Here are our responses to the comments. The comments are numbered and in italics.

Thanks for your attention and my best regards

Xiaoyu Li, Jiahong Yuan, Yan Shi, Zilai Sun and Junhu Ruan.

Point 1: The authors have written an interesting but very high-level paper on the exploding amount of research on new digital stuff in finance. I like the paper and think it should be published.

Response 1: Thanks for your confirmation on our contribution and constructive comments. We have revised our work strictly based on your suggestions.

Point 2: I do not have much to add, except being a little more concrete a couple of places might be worth it. For example around section 4.1.1 on internet banking, one could cite recent advances on understand risk fast in a fintech environment via the new fintech methodology pointed out in

Gerrard, R., Hiabu, M., Kyriakou, I., Nielsen, J.P. 2019. Communication and personal selection of pension saver's financial risk   European Journal of Operational Research

and in your section 4.1.5 on stock prediction, while the cited paper might be fun, then a more serious paper predicting stocks for the long-run - that are more important for the man-on-the-street looking for long-term pension investment - and also using machine learning type of methods can be found in

Kyriakou, I., Mousavi, P., Nielsen, J.P., Scholz, M.  2019 Forecasting benchmarks of long-term stock returns via machine learning Open Access. Annals of Operations Research

There are of course many more studies relevant for the current research. However, I also understand that there will have to be some limit to many specific cases that can be treated.

Response 2: Thanks for the suggestion. We have added the literature about the emerging trends of internet finance and analyzed it in this paper.

  In section 4.1.1, the following discussion has been added: “Besides, A new research direction is how to make financial platforms such as platform of pension products more transparent and efficient [80]”. and in section 4.1.5, the following discussion has been added: “Kyriakou et al. studied the application of machine learning in stock return research [81]”. In addition, relevant literature has been added to the end of the reference.
